# Distinct spatiotemporal patterns of cortical thinning in Alzheimer's disease-type cognitive impairment and subcortical vascular cognitive impairment

Jinhee Kim [1,2,8], Jonghoon Kim[3,4,8], Yu-hyun Park[1,5,6], Heejin Yoo[1,5,6], Jun Pyo Kim[1,5,6,7], Hyemin Jang[1,5,6,7], Hyunjin Park [3,4,9✉] & Sang Won Seo[1,5,6,7,9✉]

Previous studies on Alzheimer's disease-type cognitive impairment (ADCI) and subcortical vascular cognitive impairment (SVCI) has rarely explored spatiotemporal heterogeneity. This study aims to identify distinct spatiotemporal cortical atrophy patterns in ADCI and SVCI. 1,338 participants (713 ADCI, 208 SVCI, and 417 cognitively unimpaired elders) underwent brain magnetic resonance imaging (MRI), amyloid positron emission tomography, and neuropsychological tests. Using MRI, this study measures cortical thickness in five brain regions (medial temporal, inferior temporal, posterior medial parietal, lateral parietal, and frontal areas) and utilizes the Subtype and Stage Inference (SuStaIn) model to predict the most probable subtype and stage for each participant. SuStaIn identifies two distinct cortical thinning patterns in ADCI (medial temporal: 65.8%, diffuse: 34.2%) and SVCI (fronto-temporal: 47.1%, parietal: 52.9%) patients. The medial temporal subtype of ADCI shows a faster decline in attention, visuospatial, visual memory, and frontal/executive domains than the diffuse subtype (p-value < 0.01). However, there are no significant differences in longitudinal cognitive outcomes between the two subtypes of SVCI. Our study provides valuable insights into the distinct spatiotemporal patterns of cortical thinning in patients with ADCI and SVCI, suggesting the potential for individualized therapeutic and preventive strategies to improve clinical outcomes.

[1] Departments of Neurology, Samsung Medical Center, Sungkyunkwan University School of Medicine, Seoul, Korea. [2] Departments of Neurology, Severance Hospital, Yonsei University School of Medicine, Seoul, Korea. [3] Department of Electrical and Computer Engineering, Sungkyunkwan University, Suwon, Korea. [4] Center for Neuroscience Imaging Research, Institute for Basic Science, Suwon, Korea. [5] Research Institute for Future Medicine, Samsung Medical Center, Sungkyunkwan University of Medicine, Seoul, Korea. [6] Biostatistics and Clinical Epidemiology Center, Samsung Medical Center, Seoul, Korea. [7] Samsung Alzheimers Convergence Research Center, Samsung Medical Center, Seoul, Korea. [8] These authors contributed equally: Jinhee Kim, Jonghoon Kim. [9] These authors jointly supervised this work: Hyunjin Park, Sang Won Seo. ✉email: hyunjinp@skku.edu; sangwonseo@empal.com; sw72.seo@samsung.com

Alzheimer's disease (AD) and subcortical vascular dementia (SVaD) are two major causes of dementia[1]. In recent years, growing interest in early interventions to decelerate cognitive decline has led to the concept of mild cognitive impairment (MCI)[2–5]. In this regard, the terms AD-type cognitive impairment (ADCI) and subcortical vascular cognitive impairment (SVCI) represent the cognitive impairment continuum related to AD and cerebral small vessel disease (CSVD), respectively[6]. Pathologically, ADCI is characterized by the accumulation of amyloid beta (Aβ) plaques and neurofibrillary tangles (NFT) in the cerebral cortex, leading to neuronal loss and cortical atrophy[7]. Specifically, ADCI is characterized by cortical atrophy in the medial temporal and lateral parietal regions[8]. The distribution of cortical atrophy is strongly correlated with the clinical features of ADCI patients[9].

SVCI is characterized by ischemic changes in subcortical regions, such as the white matter or deep nuclei, caused by CSVD-related ischemia and occlusion[10]. Previous studies have shown that SVCI results in cortical thinning through various mechanisms, including secondary degeneration, concomitant AD pathologies, and cortical microinfarcts[11–13]. Specifically, a direct comparison of cortical thickness between ADCI and SVCI groups showed that frontal atrophy was predominant in SVCI patients, whereas ADCI patients exhibited atrophy mainly in the medial temporal and medial parietal areas[8,11].

Recently, several reports have shown heterogeneity in the clinical phenotypes of ADCI. Based on the initial symptoms, ADCI is divided into typical amnestic syndrome and non-amnestic syndrome[14–17]. Neuropathological and imaging studies have suggested the existence of several subtypes of ADCI based on the distribution of NFT and patterns of cortical atrophy[18]. The three subtypes identified in pathological studies are the limbic-predominant, hippocampal-sparing, and typical AD subtypes[18–20]. Previous research from our group suggested the existence of several subtypes of ADCI based on patterns of cortical thinning, including the medial temporal-dominant, parietal-dominant, and diffuse atrophy subtypes[21]. However, heterogeneity in the clinical phenotypes of SVCI has not been extensively investigated and previous studies have only focused on classifying distinct subtypes (phenotypic heterogeneity) while disregarding variability in disease stages (temporal heterogeneity)[22].

The spreading pattern of cortical atrophy in ADCI typically first involves the medial temporal area, followed by the medial parietal, lateral temporal and lateral parietal, high-order sensory association, and prefrontal areas, and finally the primary sensorimotor areas. In contrast, patients with SVCI exhibit cortical atrophy that begins in the frontal and perisylvian areas and then spreads to the lateral parietal and temporal areas[8,11]. Although this progression pattern requires longitudinal validation, the collection of longitudinal data is challenging. Furthermore, it is difficult to account for both phenotypic and temporal heterogeneity. To address these issues, an event-based model (EBM) was developed. The Subtype and Stage Inference (SuStaIn) model is a novel approach that assigns individuals to subtypes and stages by evaluating the most probable subtype and choosing the stage with the highest likelihood[23,24]. This model enables a deeper understanding of temporal and phenotypic heterogeneity. However, only a few studies have applied the SuStaIn model to degenerative diseases and these studies have been limited to non-Hispanic white (NHW) populations[25–27]. Given that the Asian population has more CSVD burdens and lower Aβ frequency than NHWs, there might be differences in the spatiotemporal patterns between Asians and NHWs.

ADCI and SVCI also share overlapping pathology in some cases: concurrent CSVD burden is more frequently observed in ADCI compared to other neurodegenerative diseases. Likewise, amyloid markers are reported to be positive in 30–53% of SVCI patients. Thus, considering that ADCI and SVCI showed different distribution patterns of brain atrophy through different mechanisms, understanding the atrophic subtypes and stages for these two diseases could provide insights into their respective pathomechanisms and aid in their differential diagnosis. For example, the pattern of cortical atrophy in SVCI patients with mixed AD pathology may reflect that of AD, potentially exhibiting medial temporal and parietal thinning.

In this study, we aimed to identify distinct spatiotemporal patterns of cortical thinning in Korean patients with ADCI and SVCI using cross-sectional neuroimaging data, and to determine whether these distinct patterns were associated with differences in cognitive outcomes over time, by analyzing longitudinal neuropsychological data. We hypothesized that there are specific spatiotemporal patterns of cortical thinning in ADCI and SVCI, and that these patterns could be associated with different clinical courses.

## Materials and methods

**Participants**. Between February 2014 and October 2020, we enrolled 1,338 participants who underwent both magnetic resonance imaging (MRI) and Aβ positron emission tomography (PET) at Samsung Medical Center. Of these, 713 were diagnosed with ADCI, 208 with SVCI, and 417 were cognitively unimpaired (CU). Within the ADCI group, there were 68 patients with preclinical AD[28], 283 with MCI due to AD[3], and 362 with dementia due to AD[29,30] proposed by the National Institute on Aging-Alzheimer's Association workgroups, all confirmed to be Aβ positive on PET. The patients with SVCI met the following criteria: (1) subjective cognitive complaints from the patients or caregivers; (2) objective cognitive decline in at least one domain of memory, language, visuospatial, or frontal function evaluated by a comprehensive neuropsychological battery; (3) significant ischemia on brain MRI, defined as periventricular white matter hyperintensities (WMH) ≥ 10 mm and deep WMH ≥ 25 mm in diameter; and (4) focal neurologic symptoms or signs. We classified patients who met the SVCI criteria as having either subcortical vascular MCI ($n = 110$) or SVaD ($n = 98$) based on their impairment in activities of daily living (ADLs)[31]. CU participants met the criteria of having a cognitively normal performance (above −1 standard deviation) on neuropsychological tests, no history of neurologic or psychiatric disorders, and Aβ negativity on PET and non-significant ischemia on brain MRI.

Before imaging, all participants underwent a series of laboratory tests, including complete blood cell count, blood chemistry, vitamin B12, folate, syphilis serology, and thyroid function tests. Individuals with structural lesions, such as territorial cerebral infarction or brain tumors, were excluded from the study.

**Standard protocol approvals, registrations, and patient consent**. Written informed consent was obtained from all participants. This study was approved by the Institutional Review Board of Samsung Medical Center. All ethical regulations relevant to human research participants were followed.

**Acquisition of brain magnetic resonance images**. We acquired standardized three-dimensional (3D) T1-weighted turbo field echo, 3D fluid-attenuated inversion recovery, and T2-weighted gradient-echo MR images from each participant at Samsung Medical Center using a 3.0-T MRI scanner (Philips 3.0 T Achieva; Philips Healthcare). The 3D T1 imaging parameters were as follows: sagittal slice thickness, 1.0 mm; over contiguous slices with 50% overlap; no gap; repetition time, 9.9 ms; echo time,

4.6 ms; flip angle, 8°; and matrix size of 240 × 240 pixels reconstructed to 480 × 480 over a field of view of 240 mm.

**Cortical thickness analysis**. The CIVET anatomical pipeline version 2.1.0 was used to process the images[32]. The pipeline includes several steps: First, the native T1-weighted images were registered to the standard Montreal Neurological Institute (MNI) template[33] and bias correction was performed using the N3 algorithm[34]. The images were then divided into white matter, gray matter, cerebrospinal fluid (CSF), and background. The Constrained Laplacian-based Automated Segmentation with Proximities algorithm was used to automatically extract the inner and outer cortical surfaces[35]. Tissue classification can be challenging because of the presence of extensive WMH on MRI. To overcome tissue classification errors caused by the presence of extensive WMH, a fluid-attenuated inversion recovery image was used to automatically define the WMH region, which was then substituted for the intensity of normal peripheral tissue on the high-resolution T1-weighted image. The cortical thickness is defined as the Euclidean distance between the linked vertices of the inner and outer surfaces[36]. It was calculated in native brain spaces because of the limitations of linear stereotaxic normalization. Intracranial volume was defined as the total volume of gray matter, white matter, and CSF and was calculated using the FMRIB Software Library[37]. It was used as a covariate when comparing cortical thickness, since it is known to reflect brain size and correlates positively with cortical thickness[38]. The cortical thickness values were spatially normalized using surface-based two-dimensional registration to compare the thicknesses of the corresponding regions among participants. Finally, we obtained the regional cortical thickness of each lobe using regions defined in the MNI152 atlas.

**Aβ PET acquisition and Aβ positivity**. All participants underwent Aβ PET as follows: 561 underwent fluorine-18-labeled ($^{18}$F)-florbetaben PET and 360 $^{18}$F-flutemetamol PET at Samsung Medical Center using a Discovery Ste PET/computed tomography (CT) scanner (GE Medical Systems, Milwaukee, WI) or Biograph mCT PET/CT scanner (Siemens Medical Solutions, Malvern, PA) in 3D scanning mode that examined 47 slices of 3.3 mm and 35 slices of 4.25 mm thickness spanning the entire brain, respectively. For $^{18}$F-florbetaben PET and $^{18}$F-flutemetamol PET, 20-minute emission PET in dynamic mode (consisting of 4 × 5 min frames) was performed 90 min after injecting approximately 300 MBq of $^{18}$F-florbetaben or 185 MBq of $^{18}$F-flutemetamol.

We defined $^{18}$F-florbetaben PET as positive if the visual assessment scored 2 or 3 on the brain Aβ plaque load scoring system[39]. $^{18}$F-flutemetamol PET was defined as positive if one of five brain regions (frontal, parietal, posterior cingulate and precuneus, striatum, and lateral temporal lobes) was positive in either hemisphere[40].

**Subtype and Stage Inference model**. We applied the SuStaIn algorithm to characterize the spatial and temporal heterogeneity of patients with ADCI and SVCI (https://github.com/ucl-pond/pySuStaIn). This approach is a data-driven clustering method used to identify disease progression patterns by inferring distinct characteristics of subtypes (i.e., a group of subjects who have a similar trajectory of biomarkers) and stages (i.e., how different from normal patients at a specific time) from cross-sectional data. The trajectory within each subtype is based on the piecewise linear z-score model, in which each biomarker reaches a particular z-score relative to a normal population, and the individual's stage is determined from the z-score[24]. The optimal number of subtypes was computed by cross-validation based on information criteria

(CVIC) and log-likelihood to verify the consistency and accuracy of the model, whereas the number of SuStaIn stages was determined by the number of biomarkers designated in the model[25].

For the SuStaIn analysis, we considered five brain areas, including the medial temporal, inferior temporal, posterior medial parietal, lateral parietal, and frontal areas, which are known to be associated with ADCI and SVCI. The mean thickness was computed for each region and adjusted for covariates, including age, sex, and education level. Initially, we assessed differences in age, sex, and education level between both groups (CU vs. ADCI and CU vs. SVCI). The corrected value was considered as the residuals derived from multivariate regression analysis between cortical thickness and age, sex, and education level. Finally, we transformed adjusted cortical thickness measurements into z-scores relative to the entire normal population (noted as w-scores)[41]. The features of the normal population were computed for the CU group. In this study, both the ADCI and SVCI groups used the same input measurements for disease progression modeling.

To plot the spatiotemporal patterns of cortical thinning, we calculated the ratio of the thickness value relative to the normal population and then observed the trajectory at the population level using BrainNet Viewer[42] (https://www.nitrc.org/projects/bnv/).

**Neuropsychological tests**. To comprehensively assess cognitive function, all participants underwent the Korean version of the mini-mental state examination (K-MMSE)[43] and the Seoul Neuropsychological Screening Battery 2$^{nd}$ edition (SNSB-II)[44]. The SNSB-II evaluates many cognitive domains, including verbal and visual memory, visuoconstructive function, language, praxis, components of Gerstmann syndrome (acalculia, agraphia, right/left disorientation, and finger agnosia), and frontal/executive functions. We have described a detailed neuropsychological assessment in our previous work[45].

**Follow-up evaluations**. A total of 595 (454 ADCI, 141 SVCI) patients had longitudinal K-MMSE results ranging from 2 to 16 time points. The study participants were examined for 5.7 ± 4.1 years retrospectively from the initial K-MMSE. From the retrospective SNSB-II data, 518 (395 ADCI, 123 SVCI) patients had at least two time points, with a mean follow-up period of 5.4 ± 3.8 years.

**Statistics and reproducibility**. For the descriptive statistics of baseline demographic characteristics, we used independent sample t-tests for continuous variables and $\chi$2 tests for dichotomous variables. To investigate the differences in the baseline cognitive measures of CU, ADCI, and SVCI, we conducted an analysis of covariance with age, sex, and education level as covariates.

To investigate the differences in cognitive trajectories across SuStaIn subgroups, we used a linear mixed-effects model, in which the interactions between the SuStaIn subgroup and time interval (SuStaIn subgroup × time) were explored to determine the influence of the SuStaIn subgroup on the rate of cognitive decline, with additional fixed effects of age, sex, education level, SuStaIn subgroup, baseline cognition, and time interval from baseline. The patients were included as random effects.

To validate the stages estimated by the SuStaIn model, we used a linear regression model to examine the correlation between K-MMSE scores and the estimated disease stage, adjusting for age, sex, and education level, in both the ADCI and SVCI groups. Using the SuStaIn model, we allocated stages to each participant within ranges of 0 to 15. Since the numbers of participants allocated to 5 or above stages were small, we categorized our participants into six separate group based on their estimated

## Table 1 Demographic variables and cognitive profiles of the study population.

| Total (n = 1340) | CU (n = 419) | ADCI (n = 713) | SVCI (n = 208) | p value ADCI vs. SVCI |
|---|---|---|---|---|
| Demographics | | | | |
| Age | 69.4 ± 8.1 | 70.8 ± 9.3[a] | 77.0 ± 7.8[a] | <0.001 |
| Sex, female (n (%)) | 252 (60.1%) | 404 (56.7%)[a] | 148 (71.2%)[a] | <0.001 |
| Education (years) | 12.1 ± 4.7 | 11.7 ± 4.7[a] | 8.7 ± 5.4[a] | <0.001 |
| APOE genotype | | | | <0.001 |
| E3 homozygotes | 297 (70.9%) | 281 (39.4%) | 137 (65.9%) | |
| E2 carrier | 41 (9.8%) | 21 (3.0%) | 16 (7.7%) | |
| E4 carrier | 81 (19.3%) | 411 (57.6%) | 55 (26.4%) | |
| Amyloid PET positivity (n (%)) | 0 | 713 (100.0%) | 71 (34.1%) | <0.001 |
| Cognitive profiles | | | | |
| K-MMSE | 28.2 ± 1.9 | 21.8 ± 6.2 | 21.8 ± 5.3 | 1.000[b] |
| SNSB-II | (n = 369) | (n = 554) | (n = 162) | |
| Digit span backward | 4.2 ± 1.4 | 3.4 ± 1.3 | 3.0 ± 1.0 | 0.304[b] |
| K-BNT | 48.9 ± 6.6 | 38.7 ± 12.9 | 35.9 ± 11.1 | 1.000[b] |
| RCFT-copy | 32.8 ± 3.5 | 25.7 ± 10.5 | 24.3 ± 9.0 | 0.888[b] |
| SVLT-delayed recall | 7.0 ± 2.2 | 1.6 ± 2.4 | 2.4 ± 2.6 | <0.001[b] |
| RCFT-delayed recall | 15.2 ± 6.4 | 4.7 ± 5.7 | 6.3 ± 6.4 | <0.001[b] |
| COWAT-Semantic | 33.5 ± 9.4 | 22.3 ± 10.0 | 19.1 ± 9.7 | 0.568[b] |
| Stroop color reading | 88.8 ± 21.1 | 51.9 ± 32.7 | 42.4 ± 26.5 | 0.312[b] |

*ADCI* Alzheimer's disease-type cognitive impairment, *SVCI* subcortical vascular cognitive impairment, *APOE* Apolipoprotein, *K-MMSE* = Korean version of the Mini-Mental State Examination, *SNSB-II* = Seoul Neuropsychological Screening Battery 2nd edition, *K-BNT* Korean version of the Boston Naming Test, *RCFT* Rey–Osterrieth Complex Figure Test, *SVLT* Seoul Verbal Learning Test, *COWAT* Controlled Oral Word Association Test.
[a]Indicates a significant statistical difference at $p < 0.05$ between CU and ADCI and CU and SVCI for age, sex and education levels.
[b]The p values were obtained by analysis of covariance after controlling for age, sex, and education level, and adjusted using Bonferroni's method because of inflated type I error.

disease stages: 0, 1, 2, 3, 4, and 5 or above. The statistical analyses were performed using STATA (version 15; StataCorp, College Station, TX, USA). P-values were corrected for multiple comparisons using the Bonferroni method, and a $p$-value < 0.05 was considered statistically significant for all analyses.

**Reporting summary**. Further information on research design is available in the Nature Portfolio Reporting Summary linked to this article.

## Results
**Clinical characteristics of participants**. Table 1 shows the demographics of the patients with CU, ADCI, and SVCI. There were differences in age, sex, and education level between group comparisons (CU vs. ADCI and CU vs. SVCI). Therefore, we considered w-scores after controlling for age, sex, and education level. Participants with SVCI were older (70.8 ± 9.3 vs. 77.0 ± 7.8 years, $p < 0.001$) and had fewer years of education (11.7 ± 4.7 vs. 8.7 ± 5.4 years, $p < 0.001$). The prevalence of females was higher ($p < 0.001$) and the frequency of Aβ PET positivity was lower ($p < 0.001$) in the SVCI group. The prevalence of apolipoprotein (APOE) 4 carriers was higher in the ADCI group, whereas the prevalence of APOE2 carriers and APOE3 homozygotes was higher in the SVCI group ($p < 0.001$).

When comparing baseline cognition, the ADCI group showed poorer memory performance (both verbal and visual memory on Seoul Verbal Learning Test [SVLT] delayed recall and Rey–Osterrieth Complex Figure Test [RCFT] delayed recall) than the SVCI group after adjusting for age, sex, and education level. There were no significant differences in K-MMSE scores or other cognitive domains between the groups.

**Identified subtypes of cortical thinning in ADCI and SVCI**. SuStaIn identified two subtypes of cortical atrophy with distinct trajectories in five brain areas in both the ADCI and SVCI groups (Fig. 1).

In the ADCI group, two distinct patterns revealed different spatiotemporal trajectories (Fig. 2A). The first subtype, denoted as the medial temporal subtype, showed spatiotemporal changes starting from the medial temporal regions and continuing to the inferior temporal, posterior medial parietal, lateral parietal, and finally frontal areas in the early stages. The second subtype, denoted as the diffuse subtype, showed scattered cortical thinning in the early stages, including in the posterior medial parietal, lateral parietal, and frontal regions. As the disease progresses from stage 1 to stage 9, affected regions gradually spread to most of disease-specific vulnerable regions. However, participants allocated to the stage 11 showed most of disease-specific vulnerable regions and after the stage 11, their cortical thinning was more prominent in those regions.

In the SVCI group, two different patterns represented differentiable spatiotemporal changes based on the moderate stage (Fig. 2B). The first subtype, the frontotemporal subtype, exhibited cortical thinning in the inferomedial temporal and frontal regions in the early stage before spreading to the posterior medial parietal and lateral parietal regions. Conversely, the second subtype, denoted as the parietal subtype, was initiated in the posterior medial and lateral parietal regions and extended to other brain regions over time.

The optimal number of subtypes was determined by using CVIC and log-likelihood with respect to varying number of subtypes (Supplementary Fig. 1). Using two subtypes had the lowest CVIC and highest log-likelihood in the SVCI patients. Using two subtypes had the lowest CVIC, but there was no statistically significant difference in log-likelihood values between using two and three subtypes for ADCI patients. Additional results of using three subtypes and the comparison with using two subtypes for the ADCI patients are reported in the supplement (Supplementary Fig. 2 and Tables 1–3).

**Subtype differences**. The demographic characteristics of each subtype are presented in Table 2. When comparing the differences between subtypes within ADCI, there were no

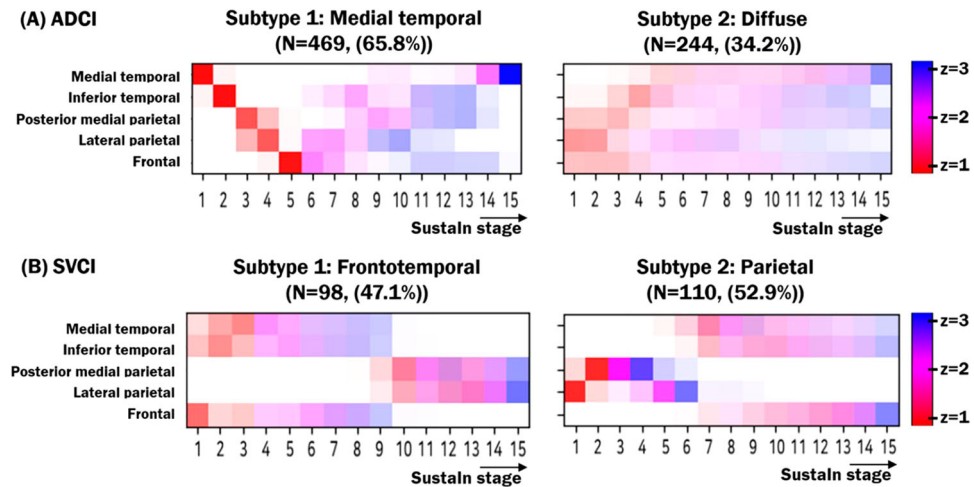

**Fig. 1 Positional variance diagrams for ADCI and SVCI subtypes.** (**A**) For the ADCI group, the disease progression predicted by SuStaIn algorithm is characterized by two distinct patterns: medial temporal and diffuse subtypes. (**B**) For the SVCI group, the two disease progression patterns represent frontotemporal subtype and parietal subtypes. Biomarker values (i.e., cortical thickness) were plotted using z-scores relative to normal controls using color mapping ranging from red (z = 1) to pink (z = 2) to and blue (z = 3). ADCI = Alzheimer's disease type cognitive impairment; SVCI = subcortical vascular cognitive impairment.

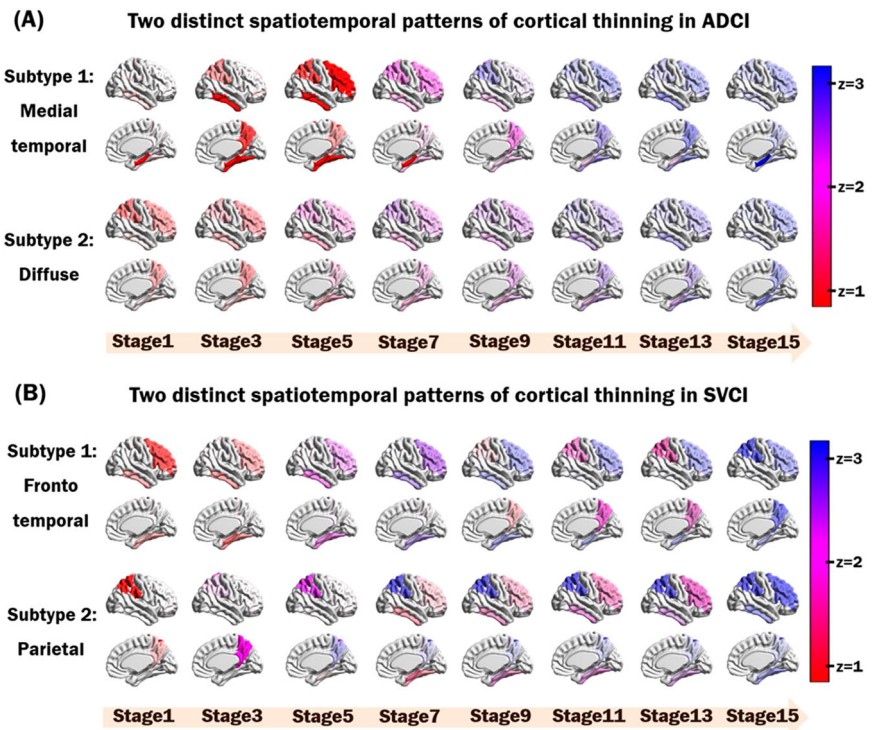

**Fig. 2 Spatiotemporal patterns of cortical thinning for distinct subtypes in both ADCI and SVCI.** Each progression pattern shows the extent of z-score change for cortical thinning relative to normal controls. (**A**) For the ADCI group, the distinct spatiotemporal transition of cortical thinning results in two subtypes: medial temporal and diffuse. (**B**) For the SVCI group, the distinct spatiotemporal transition of cortical thinning results in two subtypes: frontotemporal and parietal. Visualizations were generated using BrainNetViewer[42]. ADCI = Alzheimer's disease type cognitive impairment; SVCI = subcortical vascular cognitive impairment.

differences in demographics, APOE genotypes, and Aβ PET positivity. Compared to the diffuse subtype, the medial temporal subtype had lower SVLT-delayed recall ($p = 0.009$) and RCFT-delayed recall scores ($p = 0.018$). A comparison of demographic and cognitive characteristics between the two SVCI subtypes did not show any statistical differences across the groups.

**Differences in cognitive trajectories across SuStaIn subgroups.** In the linear mixed-effects model, which tested the interaction effect of the ADCI subtype and time on cognitive decline, the medial temporal subtype showed a steeper decrease in the K-MMSE score ($p < 0.001$), digit span backward ($p < 0.001$), Korean version of the Boston Naming Test ($p < 0.001$), RCFT copy ($p < 0.001$), RCFT-delayed recall ($p = 0.008$), Controlled

**Table 2 Baseline demographic variables and cognitive profiles of each SuStaIn subtype.**

| ADCI (n = 713) | Medial temporal (n = 469) | Diffuse (n = 244) | p value |
|---|---|---|---|
| **Demographics** | | | |
| Age | 70.3 ± 9.4 | 71.7 ± 9.2 | 0.053 |
| Female sex (n (%)) | 277 (59.1%) | 127 (52.1%) | 0.073 |
| Education (years) | 11.6 ± 4.6 | 11.8 ± 4.8 | 0.667 |
| APOE genotype | – | – | 0.922 |
| E3 homozygotes | 186 (39.7%) | 95 (38.9%) | – |
| E2 carrier | 13 (2.8%) | 8 (3.3%) | – |
| E4 carrier | 270 (57.6%) | 141 (57.8%) | – |
| **Cognitive profiles** | | | |
| K-MMSE | 21.4 ± 6.2 | 22.5 ± 6.1 | 0.196[a] |
| SNSB-II | (n = 366) | (n = 188) | – |
| Digit span backward | 3.5 ± 1.4 | 3.4 ± 1.2 | 1.000[a] |
| K-BNT | 37.9 ± 13.3 | 40.3 ± 12.1 | 0.293[a] |
| RCFT-copy | 25.7 ± 10.5 | 25.7 ± 10.6 | 1.000[a] |
| SVLT-delayed recall | 1.3 ± 2.1 | 2.0 ± 2.7 | 0.009[a] |
| RCFT-delayed recall | 4.1 ± 5.3 | 5.7 ± 6.4 | 0.018[a] |
| COWAT-Semantic | 21.8 ± 10.1 | 23.2 ± 9.9 | 1.000[a] |
| Stroop color reading | 52.0 ± 32.5 | 51.7 ± 33.2 | 1.000[a] |
| **SVCI (n = 208)** | **Frontotemporal (n = 98)** | **Parietal (n = 110)** | **p value** |
| **Demographics** | | | |
| Age | 77.2 ± 7.9 | 76.8 ± 7.7 | 0.715 |
| Female sex (n (%)) | 69 (70.4%) | 79 (71.8%) | 0.823 |
| Education (years) | 9.4 ± 5.5 | 8.1 ± 5.2 | 0.079 |
| APOE genotype | – | – | 0.544 |
| E3 homozygotes | 66 (67.4%) | 71 (64.6%) | – |
| E2 carrier | 9 (9.2%) | 7 (6.4%) | – |
| E4 carrier | 23 (23.5%) | 32 (29.1%) | – |
| Amyloid PET positivity (n (%)) | 31 (31.6%) | 40 (36.4%) | 0.473 |
| **Cognitive profiles** | | | |
| K-MMSE | 22.9 ± 4.7 | 20.9 ± 5.7 | 0.051[a] |
| SNSB-II | (n = 97) | (n = 106) | – |
| Digit span backward | 3.0 ± 1.0 | 2.7 ± 1.2 | 1.000[a] |
| K-BNT | 35.9 ± 12.2 | 33.0 ± 11.7 | 0.698[a] |
| RCFT-copy | 23.8 ± 9.9 | 21.7 ± 10.4 | 1.000[a] |
| SVLT-delayed recall | 2.7 ± 2.9 | 1.9 ± 2.4 | 0.270[a] |
| RCFT-delayed recall | 6.8 ± 6.7 | 5.4 ± 6.0 | 0.872[a] |
| COWAT-Semantic | 19.5 ± 10.3 | 16.7 ± 9.7 | 0.393[a] |
| Stroop color reading | 43.1 ± 28.7 | 37.0 ± 27.1 | 0.963[a] |

[a]The p values were obtained by analysis of covariance after controlling for age, sex, education level and adjusted using Bonferroni's method because of inflated type I error.
*ADCI* Alzheimer's disease-type cognitive impairment, *SVCI* subcortical vascular cognitive impairment, *APOE* Apolipoprotein, *K-MMSE* Korean version of the Mini-Mental State Examination, *SNSB-II* Seoul Neuropsychological Screening Battery 2nd edition; *K-BNT* Korean version of the Boston Naming Test, *RCFT* Rey–Osterrieth Complex Figure Test, *SVLT* Seoul Verbal Learning Test, *COWAT* Controlled Oral Word Association Test.

Oral Word Association Test-Semantic ($p < 0.001$), and Stroop color reading test ($p < 0.001$).

However, in the longitudinal analysis of SVCI, no neuropsychological tests showed significant differences in cognitive trajectories according to subtype (Fig. 3).

**Correlation between MMSE scores and estimated disease stages.** To evaluate the clinical relevance of these model-estimated stages, we utilized K-MMSE score data. Upon comparing the K-MMSE scores across these groups, we observed a consistent trend of decline as the stage advanced in both ADCI and SVCI groups (ADCI, *p* for trend $< 0.001$; SVCI, *p* for trend $= 0.006$; Fig. 4).

**Discussion**

In the present study, we investigated the distinct spatiotemporal patterns of cortical thinning in carefully phenotyped participants with ADCI and SVCI who underwent MRI and Aβ PET. Our major findings are as follows: First, both ADCI and SVCI showed distinct spatiotemporal patterns of cortical thinning. Second, the

subtypes of ADCI revealed different clinical outcomes according to the spatiotemporal patterns. Taken together, our findings provide a more insightful understanding of the distinct progression of the spatiotemporal patterns of cortical thinning in participants with ADCI and SVCI. Furthermore, our results may help in designing individualized therapeutics and preventive strategies to improve clinical outcomes.

Our first major finding was that ADCI and SVCI showed distinct spatiotemporal patterns of cortical thinning. Specifically, the SuStaIn model identified two topographical subtypes of cortical atrophy in ADCI, which we termed "medial temporal" and "diffuse" subtypes according to the earliest regions of cortical thinning. In the medial temporal subtype, cortical atrophy began in the medial temporal and inferior temporal regions, followed by the medial parietal and lateral parietal regions. For the diffuse subtype, as the name suggests, cortical thinning progressed in the entire area without initial focal atrophy. The spatiotemporal patterns of the medial temporal subtype were consistent with the typical progression of AD observed in previous studies. Pathological studies have shown that NFT preferentially affects the medial temporal area in MCI and the earlier stages of AD, and

**Table 3 Mixed effects models analyzing the relationship between SuStaIn subtype and rate of cognitive decline.**

| Cognition parameter | ADCI subtype × time | | SVCI subtype × time | |
|---|---|---|---|---|
| | β (S.E) | *p* for interaction | β (S.E) | *p* for interaction |
| **K-MMSE** | 0.469 (0.07) | <0.001[a] | −0.153 (0.06) | 0.144[a] |
| **SNSB-II** | | | | |
| Digit span backward | 0.082 (0.02) | <0.001[a] | −0.027 (0.02) | 0.896[a] |
| K-BNT | 0.897 (0.13) | <0.001[a] | −0.243 (0.11) | 0.176[a] |
| RCFT-copy | 0.482 (0.12) | <0.001[a] | 0.149 (0.13) | 1.000[a] |
| SVLT-delayed recall | −0.006 (0.04) | 1.000[a] | 0.007 (0.04) | 1.000[a] |
| RCFT-delayed recall | 0.319 (0.08) | <0.001[a] | 0.153 (0.09) | 0.856[a] |
| COWAT-Semantic | 1.018 (0.14) | <0.001[a] | −0.112 (0.13) | 1.000[a] |
| Stroop color reading | 1.979 (0.37) | <0.001[a] | −0.342 (0.39) | 1.000[a] |

Model: fixed effect: age, sex, education level, SuStaIn subtype, baseline cognition, time interval from initial evaluation (years), sustain subtype × time interval; random effect: subject.
[a]*p* values were adjusted using Bonferroni's method because of inflated type I error.
*ADCI* Alzheimer's disease-type cognitive impairment, *SVCI* subcortical vascular cognitive impairment, *K-MMSE* Korean version of the Mini-Mental State Examination, *SNSB-II* Seoul Neuropsychological Screening Battery 2nd edition, *K-BNT* Korean version of the Boston Naming Test, *RCFT* Rey–Osterrieth Complex Figure Test, *SVLT* Seoul Verbal Learning Test, *COWAT* Controlled Oral Word Association Test.

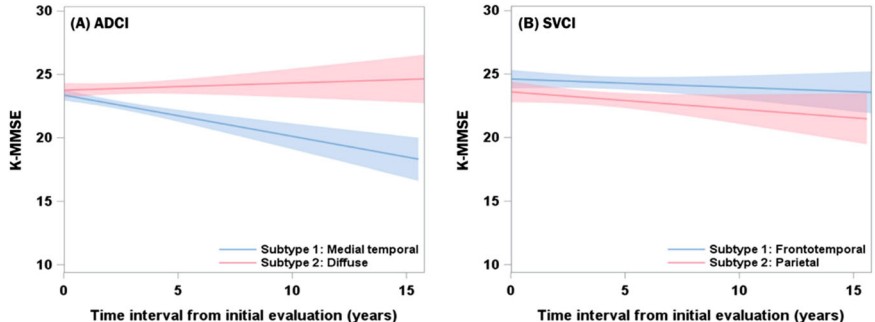

**Fig. 3 Longitudinal trajectory of K-MMSE in patients with ADCI and SVCI.** Each line represents estimated cognitive trajectories with 95% confidence intervals. (**A**) The medial temporal subtype in ADCI patients (blue line) showed a faster cognitive decline than the diffuse subtype (red line). (**B**) On the other hand, in SVCI patients, there was no significant difference in cognitive trajectory between the two groups. Number of patients; ADCI, N = 454; SVCI, N = 141. K-MMSE = Korean version of the mini-mental state examination; ADCI = Alzheimer's disease type cognitive impairment; SVCI = subcortical vascular cognitive impairment.

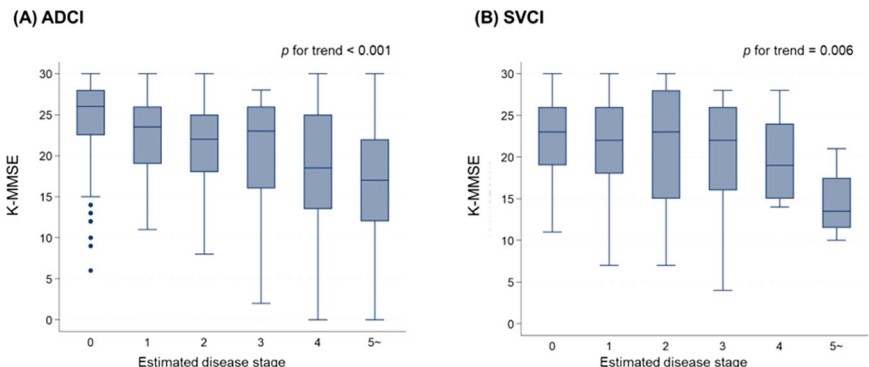

**Fig. 4 Correlation between K-MMSE scores and estimated disease stages in ADCI and SVCI groups.** In both (**A**) ADCI and (**B**) SVCI groups, as the estimated disease stage progresses, a decline in K-MMSE scores is observed. Each box plot illustrates the score distribution: horizontal lines within boxes indicate median values, boxes represent the range from the 25th to the 75th percentile, vertical extending lines denote adjacent values, representing the most extreme values within 1.5 times the interquartile range from the 25th and 75th percentiles, and dots show outlier observations. Number of patients; ADCI, N = 713; SVCI, N = 208. K-MMSE = Korean version of the mini-mental state examination; ADCI = Alzheimer's disease-type cognitive impairment; SVCI = subcortical vascular cognitive impairment.

spreads to the entire cortex as AD progresses from earlier to later stages[7,46]. The patterns in the medial temporal subtype are also in accordance with neuroimaging studies that showed differences in medial temporal cortical thickness between individuals with amnestic MCI and healthy elderly individuals and in the more diffuse isocortical areas between amnestic MCI and ADD (Alzheimer's disease dementia)[47]. The diffuse subtype, on the other hand, shows a trajectory without a predilection for the medial

temporal area and might correspond to atypical AD. In a pathologic study of atypical AD, the distribution of NFT differed according to each focal cortical syndrome, and unlike typical AD, a medial temporal dominant distribution was not observed[48]. A previous study that divided participants with ADCI into subtypes using SuStaIn analysis suggested three subtypes using 13 regional MRI volumes: typical, cortical, and subcortical subtypes[25]. Considering that in the typical subtype, atrophy starts in the hippocampus and amygdala, and in the cortical subtype, atrophy predominantly affects cortical regions without any predilection for specific cortical areas, the typical and cortical subtypes correspond to the medial temporal and diffuse subtypes, respectively.

Our noteworthy finding was that participants with SVCI were also classified into two subtypes showing distinct spatiotemporal patterns of cortical thinning: frontotemporal and parietal subtypes. In the frontotemporal subtype, cortical atrophy begins in the frontal and inferomedial temporal regions, and spreads to the posterior medial and lateral parietal regions. In the parietal subtype, cortical atrophy occurs in reverse order, starting in the posterior medial and lateral parietal regions and spreading to the frontal and inferomedial temporal regions. To our knowledge, distinct spatiotemporal patterns of cortical thinning in patients with SVCI have not been extensively investigated. The frontotemporal subtype is consistent with the known progression pattern of SVCI, in that cortical thinning starts in the frontal and temporal areas and spreads to the parietal area[8,11]. However, we hypothesized that the parietal subtype might be related to SVCI in participants with mixed AD pathologies, because the parietal area is known to be more vulnerable to AD[8,11]. However, there was no difference in Aβ positivity between the two subtypes. Considering that only 25% of Aβ-positive SVCI participants showed tau accumulation, further studies comparing tau positivity between the two subtypes are necessary because tau more directly represents cortical atrophy than Aβ does[49]. Our results are the first to show distinct spatiotemporal patterns of cortical thinning in participants with SVCI. Beyond our subtype classifications, it's crucial to note that the disease stages predicted by the SuStaIn model showed a significant correlation with K-MMSE scores both in ADCI and SVCI group. This additional finding suggests the clinical relevance of the SuStaIn model, which considers both spatial and temporal heterogeneity.

Another major finding was that the subtypes of ADCI revealed different clinical outcomes according to spatiotemporal patterns. The medial temporal subtype of ADCI showed a more rapid decline in all cognitive domains except for verbal memory. Our findings are consistent with those of a previous study showing that the risk of conversion from MCI to ADD is highest in the typical subtype[25]. In the comparison of clinical outcomes in SVCI, the frontotemporal subtype showed faster deterioration in language function; however, the difference was not statistically significant. We expected that there would be a difference in the rate of cognitive decline or the main impaired cognitive domain according to spatial patterns, but the results were different from our expectations.

The strength of this study is that we consecutively recruited and followed up carefully phenotyped participants with ADCI and SVCI who had multimodal imaging markers. However, our study has some limitations. First, it was difficult to distinguish the order of events occurring at lower frequencies, even when we attempted to predict the trajectories using the SuStaIn model. Therefore, in future longitudinal studies, it will be necessary to determine participants' temporal relationships. Second, we were unable to consider the effects of other neurodegenerative pathologies, including NFT, α-synuclein, transactive response deoxyribonucleic acid-binding protein,

argyrophilic grain pathology, and hippocampal sclerosis, because we performed autopsies in only a few cases. We selected the five sub-parcellations based on prior studies and established clinical knowledge, prioritizing brain areas of notable clinical relevance and importance. However, given these sub-parcellations are so large, there is large physiological variance even within each sub-parcellation. Additionally, we observed an increasing trend of K-MMSE scores in the diffuse subtype of ADCI. This trend may reflect differences in the distribution of cognitive stages in each subtype. Notably, the diffuse subtype exhibits a significantly higher frequency of preclinical AD (Aβ-positive CU individuals) compared to the medial temporal subtype (18.6% vs. 7.6%, $p = 0.001$). This would be an important consideration, as previous studies investigating the trajectories of preclinical AD have shown that only 13.5%–22.9% of these patients progress to cognitive decline[50–52]. One study reported that 65.5% of preclinical AD patients either maintain stable cognitive functions or even experience a slight improvement[53]. Finally, our study population included a large proportion of patients with cognitive impairment, which may limit the generalizability of our findings to other populations. Nevertheless, our study is worthwhile in that it is the first to predict heterogeneous trajectories of cortical thinning in well-characterized ADCI and SVCI in non-NHW cohorts.

In conclusion, our findings suggest distinct spatiotemporal patterns of cortical thinning in participants with ADCI and SVCI. Furthermore, distinct spatiotemporal patterns may affect the cognitive trajectories in ADCI. Therefore, our results may motivate clinicians to identify distinct subtypes of ADCI and SVCI, which may help predict the prognosis of ADCI and SVCI based on initial imaging markers.

## Data availability

The data that support the findings of this study are not openly available due to patient confidentiality. Data can only be shared with the approval of the Samsung Medical Center's Institutional Review Board upon reasonable request.

## Code availability

The codes for SusTaIn analysis are available at https://github.com/ucl-pond/pySuStaIn, for representing visualization at https://www.nitrc.org/projects/bnv/

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

## Acknowledgements

This research was supported by the National Research Foundation (NRF-2020M3E5D2A01084892 and NRF-2021R1I1A1A01061044), Institute for Basic Science (IBS-R015-D1), Ministry of Science and ICT (IITP-2020-2018-0-01798), AI Graduate School Support Program (2019-0-00421), ICT Creative Consilience program (IITP-2020-0-01821), and Artificial Intelligence Innovation Hub program (2021-0-02068).

## Author contributions

J.K.: data collection, data analysis, and writing original draft. J.K.: data analysis and writing original draft. Y.P.: data processing. H.Y.: data analysis and visualization. J.P.K.: project administration. H.J.: project administration. H.P.: conceptualization, supervision, review, and editing. S.W.S.: conceptualization, supervision, review, and editing.

## Competing interests

The authors declare no competing interests.

## Additional information

**Peer review information** : *Communications Biology* thanks Marlene Tahedl and the other, anonymous, reviewer(s) for their contribution to the peer review of this work. Primary Handling Editors: David Owen, Gene Chong and Christina Karlsson Rosenthal. A peer review file is available.

