## [Peer review file · Communications Biology]

Reviewers' comments:

Reviewer #1 (Remarks to the Author):

This paper investigates the heterogeneous atrophy progression patterns of Alzheimer's disease-type cognitive impairment (ADCI) and subcortical vascular cognitive impairment (SVC) using the Sustain model. The study is particularly interesting because there is a lack of research on the heterogeneity of SVC, and exploring this aspect in the Asian population adds significant value.

- One major concern is the validation of the proposed model. In the studies of heterogeneity like this, determining the optimal number of disease subtypes is crucial. However, the paper lacks sufficient details about the cross-validation procedure for determining the optimal number of clusters. It is essential to provide additional information about the metrics used and how the results varied based on the number of subtypes, etc. The current reports do not adequately demonstrate the selection process and results.

- To further validate the Sustain model, additional experiments are suggested. For example, evaluating whether the estimated stages of the model accurately reflect the actual progression of each disease by examining cognitive scores such as MMSE over time would be beneficial. Validation using such longitudinal data might improve the value of the manuscript.

- Figure 3 displays an intriguing observation, where the longitudinal MMSE of the diffuse subtype appears to increase over time. This finding contradicts the typical progression of ADCI patients, where cognition generally worsens over time. It would be essential to provide statistics such as slope and correlation for each subtype and offer plausible explanations for this unexpected result.

- Furthermore, the paper indicates that the spatiotemporal patterns of the two ADCI subtypes display distinct differences in the early stages (Stage 1-5), but seem to converge after Stage 11. The authors should elaborate on their thoughts about the convergence of atrophy patterns over time and discuss potential reasons for this phenomenon.

In summary, this paper presents intriguing findings on the heterogeneous progression patterns of ADCI and SVC. However, it would greatly benefit from addressing the concerns raised, particularly by providing more details on the validation of the Sustain model. Further clarification and elaboration in these areas would enhance the overall strength and credibility of the study's conclusions.

Reviewer #2 (Remarks to the Author):

Summary

Dear Editor,

Thank you for the opportunity to review the manuscript "Distinct spatiotemporal patterns of cortical thinning in Alzheimer's disease-type cognitive impairment and subcortical vascular cognitive impairment".

In this study, the authors aim to identify distinct subtypes of two disorders of cognitive impairment, namely Alzheimer's disease-type cognitive impairment (ADCI) and subcortical vascular cognitive impairment (SVC), based on differential spatiotemporal patterns of cortical thinning. Moreover, they aim to match the distinct subtypes to differential patterns of cognitive decline. They investigate cross-sectional T1-weighted magnetic resonance imaging (MRI) data from 713 ADCI, 208 SVC and 417 cognitively unimpaired (CU) Korean subjects. Moreover, rich neuropsychological data including several follow-up assessments is available for the patients. To classify distinct subtypes of cortical thinning, they analyze averaged cortical thickness (CT) of five coarse brain regions (including medial temporal, inferior temporal, posterior medial parietal, lateral parietal, and frontal areas). These CT values are first normalized with respect to a control population, by calculating z-scores for each individual patient and brain region with respect to the CU group. Then, the z-scored patient data are input into a pre-trained model ("SuStaIn") to identify CT-based subtypes for the ADCI and SVC group separately. For both conditions, two

subtypes were identified (ADCI: "medial temporal" and "diffuse"; SVCI: "frontotemporal" and "parietal". Furthermore, the medial temporal ADCI subgroup exhibited pronounced cognitive decline in visuospatial cognitive tests over time; no other correspondences between neuroimaging subtypes and longitudinal neuropsychological progression were identified.

Comments to authors

Positive:

- The study addresses an important problem in the neuroimaging literature of cognitive impairment (CI): Indeed, spatiotemporal patterns of CT are largely neglected in the literature, which mainly focuses on phenotypic differences. However, by neglecting the factor "time" when analyzing CT, invaluable diagnostic information is lost, which might in turn have implications for therapeutic treatment. Therefore, the study question of the authors is well-justified.
- Introduction: The authors motivate their study well and cite adequate literature which elaborate on the importance of the research question, as briefly outlined in the above bullet point.
- The study profits from a rich neuropsychological data set with longitudinal follow-up assessments
- In terms of language, the manuscript is well written.
- The figures are clear and adequate.
- In statistical testing, adequate models are applied and the most important confounding variables in the CT literature (age, sex, educations) are corrected for.

Major issues:

- The way the manuscript is written at the moment, it is not clear if the authors provide the analysis adequate to answer their research question: The claim is to identify spatiotemporal patterns of cortical thinning. However, it seems from the manuscript that there is no longitudinal MRI data available. The lack of longitudinal MRI data, in turn, makes identification of spatiotemporal brain morphometric measures impossible on an individual level. However, it may be possible on a group-level, if e.g., age between subjects is taken as a substitute for individual ageing. However, then one would need to correct also carefully for physiological effects of ageing, which is also not communicated (cf. also next bullet point). However, it is not clear in the manuscript if and how any of these procedures were performed. The authors do not communicate their classification procedure sufficiently to reconstruct the methodological approach: Instead, it is referred to the original publication of the model they used for this classification ("SuStaIn"). However, I believe it is imperative to at least outline the rough mechanisms of that model: at minimum, the reader should be able to comprehend how that model claims to differentiate spatiotemporal CT patterns without having longitudinal MR data at hand.
- It is not clear how the z-scores for were exactly calculated. Did the authors use the entire CU group as reference for each single subject? If so, then how did the authors correct for physiological effects of ageing (cf., e.g., Bethlehem, Seidlitz et al., 2022, Nature: "Brain Charts for the Human Lifespan")? If, however, for a given patient only a subset of age-/sex-matched patients was taken as a reference, this would provide a relatively easy opportunity to correct for physiological effects of ageing (cf., e.g. Tahedl et al., 2021, Journal of Neurology: "Cortical progression patterns in individual ALS patients across multiple timepoints: a mosaic-based approach for clinical use").

Minor issues:

- It is not fully clear why the authors chose the five cortical regions which they investigated. The provided motivation ("affected in CI") is not convincing or at least needs further elucidation, since taken together, the five sub-parcellations almost sum up to the entire cortex. Moreover, given these sub-parcellations are so large, there is large physiological variance even within each sub-parcellation. Please motivate why such large sub-parcels were investigated and discuss the choice of parcellation-size.
- Moreover, the choice of the specific patient subgroups (ADCI vs. SVCI) is not fully clear. Please elaborate on the benefit of investigating them next to each other in one study.
- Table 1 does not list demographic details of the CU population. Please provide those since that information is essential to understand whether the control population is an adequate reference for the patient populations.
- Acquisition of MRI data (ll 114): The authors state to have acquired a "standardized T2-weighted, three-dimensional (3D) T1-weighted turbo field echo, three-dimensional fluid-attenuated inversion recovery" > here, there might be typo, ("T2-weighted [...] T1-w image"?)

- LI 174: The authors state to have corrected CT for "age, gender and education". It is not specified how the correction procedure was performed. However, this information is crucial to track the adequacy of the analysis for the study question (see comments above)
- L 212: a "two-tailed p-value of 0.05" was considered statistically significant > this is a little confusing since usually, a two-tailed p-value is significant below $0.05/2 = 0.025$. Please comment/specify.
- L 255: "Interactive effect" should read "interaction effect"
- LI 306: Here, in the middle of the discussion, a new hypothesis appears ("we hypothesized that the parietal subtype might be related to SVCI in participants with mixed AD pathologies, because the parietal area is known to be more vulnerable to AD") – which had never been articulated before in the manuscript. Please avoid confronting the reader with new hypothesis in the discussion. Either delete this discussion section, or preferable include this as a hypothesis into your introduction and specify methods / results at the adequate sections in the manuscript
- Tables, minor suggestion: the asterisk symbol ("*") is usually associated with emphasizing significant p-values. Here, you use it to specify how p-values were obtained. It could be helpful to some reader to use a different symbol or footnote and use the asterisk to emphasize p-value-significance instead

Response to the Reviewers

REVIEWER 1

1. One major concern is the validation of the proposed model. In the studies of heterogeneity like this, determining the optimal number of disease subtypes is crucial. However, the paper lacks sufficient details about the cross-validation procedure for determining the optimal number of clusters. It is essential to provide additional information about the metrics used and how the results varied based on the number of subtypes, etc. The current reports do not adequately demonstrate the selection process and results.

Response: Sorry for the lack of detail. As you pointed out, the optimal number of subtypes is an important factor. Our study used 10-fold cross-validation to identify the optimal number of subtypes. This procedure evaluated the cross-validation information criterion (CVIC) and log-likelihood with respect to varying number of subtypes from one to four. As shown in the figure below (Supplementary Fig. 1), using two subtypes led to the lowest CVIC and highest log likelihood for SVCI patients. For ADCI patients, using two subtypes had slightly higher log-likelihood (mean value -952.96) than using just one subtype (mean value -954.20) and using two subtypes had the lowest CVIC. However, using one subtype leads to not performing the subtype analysis, and thus using two subtypes for ADCI has merit. Details were added to the Methods, Results, and Supplementary sections.

Supplementary Fig.1. Plots of log-likelihood and cross-validation information criterion (CVIC) with respect to different numbers of subtypes. The top row is for the ADCI patients and the bottom row is for the SCVI patients. Using two subtypes was optimal due to the highest log-likelihood and lowest CVIC values for SCVI patients. Magenta line indicates mean value for log-likelihood for 10-fold cross-validation. For ADCI patients, using two subtypes had slightly higher log-likelihood (mean value -952.56) than using just one subtype (mean value -954.20) and using two subtypes had the lowest CVIC.

In the methods sections,

2.6 Subtype and Stage Inference (SuStaIn) model

“...The optimal number of subtypes was computed by cross-validation based on information criteria (CVIC) and log-likelihood to verify the consistency and accuracy of the model, whereas the number of SuStaIn stages was determined by the number of biomarkers designated in the model.”

In the results sections,

3.2 Identified subtypes of cortical thinning in ADCI and SVCI

“...The optimal number of subtypes was determined by using CVIC and log-likelihood with respect to varying number of subtypes (Supplementary Fig.1). Using two subtypes had the lowest CVIC and highest log-likelihood in both ADCI and SVCI groups.”

2. To further validate the Sustain model, additional experiments are suggested. For example, evaluating whether the estimated stages of the model accurately reflect the actual progression of each disease by examining cognitive scores such as MMSE over time would be beneficial. Validation using such longitudinal data might improve the value of the manuscript.

Response: In response to the reviewer’s comments, we have conducted additional analyses comparing the MMSE scores across the estimated stages of the model. We observed a trend of lower MMSE scores in the advanced stages of both the ADCI (p for trend < 0.001) and SVCI (p for trend = 0.006) groups, reinforcing the model's predictive validity.

Figure 4. Correlation between K-MMSE scores and estimated disease stages in ADCI (A) and SVCI (B) groups.

K-MMSE = Korean version of the mini-mental state examination; ADCI = Alzheimer’s disease-type cognitive impairment; SVCI = subcortical vascular cognitive impairment

Therefore, we have revised our manuscript as follows:

In the methods section,

2.9 Statistical analyses

“... To validate the stages estimated by the SuStaIn model, we used a linear regression model to examine the correlation between K-MMSE scores and the estimated disease stage, adjusting for age, sex, and education level, in both the ADCI and SVCI groups. Using the SuStaIn model, we allocated stages to each participant within ranges of 0 to 15 for both groups. Since the numbers of participants allocated to 5 or above stages were small, we categorized our participants into six separate groups based on their estimated disease stages: 0, 1, 2, 3, 4, and 5 or above.”

In the result section,

3.5 Correlation between MMSE scores and estimated disease stages

“To evaluate the clinical relevance of these model-estimated stages, we utilized K-MMSE score data. Upon comparing the K-MMSE scores across these groups, we observed a consistent trend of decline as the stage advanced in both ADCI and SVCI groups (ADCI, p for trend < 0.001; SVCI, p for trend = 0.006) (Figure 4).”

Figure 4. Correlation between K-MMSE scores and estimated disease stages in ADCI (A) and SVCI (B) groups.

K-MMSE = Korean version of the mini-mental state examination; ADCI = Alzheimer’s disease-type cognitive impairment; SVCI = subcortical vascular cognitive impairment

In the discussion section,

“... Our results are the first to show distinct spatiotemporal patterns of cortical thinning in participants with SVCI. Beyond our subtype classifications, it's crucial to note that the

disease stages estimated by the SuStaIn model showed a significant correlation with K-MMSE scores both in ADCI and SVCI groups. This additional finding suggests the clinical relevance of the SuStaIn model, which considers both spatial and temporal heterogeneity.”

3. Figure 3 displays an intriguing observation, where the longitudinal MMSE of the diffuse subtype appears to increase over time. This finding contradicts the typical progression of ADCI patients, where cognition generally worsens over time. It would be essential to provide statistics such as slope and correlation for each subtype and offer plausible explanations for this unexpected result.

Response: Thank you for pointing out the intriguing observation in Figure 3. Upon revisiting our data, we identified errors in the records of two patients within the diffuse subtype. Consequently, we have re-analyzed the data, resulting in an updated table and figure.

Table 3. Mixed effects models analyzing the relationship between SuStaIn subtype and rate of cognitive decline

Cognition parameter	ADCI subtype × time		SVCI subtype × time	
	β (S.E)	p for interaction	β (S.E)	p for interaction
K-MMSE	0.469 (0.07)	<0.001 [†]	-0.153 (0.06)	0.144 [†]
SNSB-II				
Digit span backward	0.082 (0.02)	< 0.001 [†]	-0.027 (0.02)	0.896 [†]
K-BNT	0.897 (0.13)	< 0.001 [†]	-0.243 (0.11)	0.176 [†]
RCFT-copy	0.482 (0.12)	< 0.001 [†]	0.149 (0.13)	1.000 [†]
SVLT-delayed recall	-0.006 (0.04)	1.000 [†]	0.007 (0.04)	1.000 [†]
RCFT-delayed recall	0.319 (0.08)	<0.001 [†]	0.153 (0.09)	0.856 [†]
COWAT-Semantic	1.018 (0.14)	< 0.001 [†]	-0.112 (0.13)	1.000 [†]
Stroop color reading	1.979 (0.37)	< 0.001 [†]	-0.342 (0.39)	1.000 [†]

Model: fixed effect: age, sex, education level, SuStaIn subtype, baseline cognition, time interval from initial evaluation (years), sustain subtype × time interval; random effect: subject.

[†]p values were adjusted using Bonferroni's method because of inflated type I error.

ADCI = Alzheimer's disease-type cognitive impairment; SVCI = subcortical vascular cognitive impairment; K-MMSE = Korean version of the Mini-Mental State Examination; SNSB-II = Seoul Neuropsychological Screening Battery 2nd edition; K-BNT = Korean version of the Boston Naming Test; RCFT = Rey–Osterrieth Complex Figure Test; SVLT = Seoul Verbal Learning Test; COWAT = Controlled Oral Word Association Test

Figure 3. Longitudinal trajectory of K-MMSE in patients with ADCI and SVCI. (A) The medial temporal subtype in ADCI patients showed a faster cognitive decline than the diffuse subtype. (B) On the other hand, in SVCI patients, there was no significant difference in cognitive trajectory between the two groups.

K-MMSE = Korean version of the mini-mental state examination; ADCI = Alzheimer’s disease type cognitive impairment; SVCI = subcortical vascular cognitive impairment

Nevertheless, even after the corrections, the revised analysis still shows a trend of increasing K-MMSE scores over time in the diffuse subtype. It seems that patients in this group had better cognitive function. With the K-MMSE being repeated at relatively short intervals, it seems there might have been a learning effect in these patients. We have acknowledged this potential issue and added it to the limitations section in our discussion as follows:

In the discussion section,

“... We selected the five sub-parcellations based on prior studies and established clinical knowledge, prioritizing brain areas of notable clinical relevance and importance. However, given these sub-parcellations are so large, there is a large physiological variance even within each sub-parcellation. Additionally, the potential learning effect suspected in the diffuse subtype, due to the repetition of K-MMSE tests at relatively short intervals, might have affected the accuracy of estimating cognitive trajectory¹.”

4. Furthermore, the paper indicates that the spatiotemporal patterns of the two ADCI subtypes display distinct differences in the early stages (Stage 1-5), but seem to converge after Stage 11. The authors should elaborate on their thoughts about the convergence of atrophy patterns over time and discuss potential reasons for this phenomenon.

Response: We totally agree with the reviewer’s comments. As the disease progresses from stage 1 to stage 9, affected regions gradually spread to most of the disease-specific vulnerable regions. However, participants allocated to stage 11 showed the maximum spatial extent of the disease-specific vulnerable regions and after stage 11, their cortical thinning became

prominent in those regions.

In response to the reviewer's comments, we addressed the issues in our revised manuscript as follows.

In the result section,

3.2 Identified subtypes of cortical thinning in ADCI and SVCI

“...As the disease progresses from stage 1 to stage 9, affected regions gradually spread to most of disease-specific vulnerable regions. However, participants allocated to the stage 11 showed most of disease-specific vulnerable regions and after the stage 11, their cortical thinning was more prominent in those regions.”

REVIEWER 2

1. The way the manuscript is written at the moment, it is not clear if the authors provide the analysis adequate to answer their research question: The claim is to identify spatiotemporal patterns of cortical thinning. However, it seems from the manuscript that there is no longitudinal MRI data available. The lack of longitudinal MRI data, in turn, makes identification of spatiotemporal brain morphometric measures impossible on an individual level. However, it may be possible on a group-level, if e.g., age between subjects is taken as a substitute for individual ageing. However, then one would need to correct also carefully for physiological effects of ageing, which is also not communicated (cf. also next bullet point). However, it is not clear in the manuscript if and how any of these procedures were performed. The authors do not communicate their classification procedure sufficiently to reconstruct the methodological approach: Instead, it is referred to the original publication of the model they used for this classification (“SuStaIn”). However, I believe it is imperative to at least outline the rough mechanisms of that model: at minimum, the reader should be able to comprehend how that model claims to differentiate spatiotemporal CT patterns without having longitudinal MR data at hand.

Response: Sorry for the confusion. SuStaIn is a data-driven clustering method that allows each subject to be mapped to a distinct subtype and stage only using cross-sectional data. Each subject is mapped to a distinct subtype defined by subjects who have a similar trajectory of biomarkers and a distinct stage defined by how different the biomarker is from normal controls. Here, we used the cortical thickness as the biomarker. Thus, this approach allows longitudinal-like analysis using cross-sectional data. We have expanded the Methods sections to describe the SuStaIn method.

Our study calculated the w-scores (i.e., z-scores adjusted for co-variates) of cortical thickness². We adjusted for co-variates of age, sex, and education level. This approach corrected physiological effects by adjusting the cortical thickness based on clinical variables showing significant differences between group comparisons (CU vs. ADCI and CU vs. SVCI). We updated the texts as follows.

In the method section,

“The mean thickness was computed for each region and adjusted for co-variates including age, sex, and education level. Initially, we assessed differences in age, sex, and education level between both groups (CU vs. ADCI and CU vs. SVCI). The corrected value was considered as the residuals derived from multivariate regression analysis between cortical thickness and age, sex, and education level. Finally, we transformed adjusted cortical thickness measurements into z-scores relative to the entire normal population (noted as w-scores)⁴¹.”

2. It is not clear how the z-scores for were exactly calculated. Did the authors use the entire CU group as reference for each single subject? If so, then how did the authors correct for

physiological effects of ageing (cf., e.g., Bethlehem, Seidlitz et al., 2022, Nature: “Brain Charts for the Human Lifespan”)? If, however, for a given patient only a subset of age-/sex-matched patients was taken as a reference, this would provide a relatively easy opportunity to correct for physiological effects of ageing (cf., e.g. Tahedl et al., 2021, Journal of Neurology: “Cortical progression patterns in individual ALS patients across multiple timepoints: a mosaic-based approach for clinical use”).

Response: Thank you for your comments. As we mentioned in the previous response, we adjusted for co-variables of age, sex, and education level when computing the cortical thickness. The entire CU group was used to normalize the thickness value. We updated the texts as follows.

In the method section,

“...The mean thickness was computed for each region and adjusted for co-variables including age, sex, and education level. Initially, we assessed differences in age, sex, and education level between both groups (CU vs. ADCI and CU vs. SVCI). The corrected value was considered as the residuals derived from multivariate regression analysis between cortical thickness and age, sex, and education level. Finally, we transformed adjusted cortical thickness measurements into z-scores relative to the entire normal population (noted as w-scores)⁴¹.”

3. It is not fully clear why the authors chose the five cortical regions which they investigated. The provided motivation (“affected in CI”) is not convincing or at least needs further elucidation, since taken together, the five sub-parcellations almost sum up to the entire cortex. Moreover, given these sub-parcellations are so large, there is large physiological variance even within each sub-parcellation. Please motivate why such large sub-parcels were investigated and discuss the choice of parcellation-size.

Response: We selected the five sub-parcellations (medial temporal, inferior temporal, medial parietal, lateral parietal, and frontal regions) based on prior studies and established clinical knowledge, prioritizing brain areas of notable clinical relevance and importance. Specifically, previous studies from our group suggested that in ADCI, temporoparietal atrophy is notably pronounced, especially most prominently in the medial temporal and lateral parietal regions³. Previously, we also found that there were several subtypes of ADCI based on patterns of cortical thinning, including the medial temporal-dominant, parietal-dominant, and diffuse atrophy subtypes⁴. Given this significance, we further subdivided the temporoparietal region into four specific areas - medial temporal, inferior temporal, posterior medial parietal, and lateral parietal - to provide a more detailed analysis. However, we did not further parcellate these regions because these subregions are likely to be connected to one another and possibly involved simultaneously. The frontal lobe is an area where atrophy is evident from the early stages of SVCI and also clearly demonstrates the differences between ADCI and SVCI^{3,5}. In response to the reviewer’s comments, we addressed this issue in our revised manuscript as follows.

In the introduction section,

“...Pathologically, ADCI is characterized by the accumulation of amyloid beta (A β) plaques and neurofibrillary tangles (NFT) in the cerebral cortex, leading to neuronal loss and cortical atrophy. Specifically, ADCI is characterized by cortical atrophy in the medial temporal and lateral parietal regions³. The distribution of cortical atrophy is strongly correlated with the clinical features of ADCI patients.

SVCI is characterized by ischemic changes in subcortical regions, such as the white matter or deep nuclei, caused by cerebral small-vessel disease-related ischemia and occlusion. Previous studies have shown that SVCI results in cortical thinning through various mechanisms, including secondary degeneration, concomitant AD pathologies, and cortical microinfarcts. Specifically, a direct comparison of cortical thickness between ADCI and SVCI groups showed that frontal atrophy was predominant in SVCI patients, whereas ADCI patients exhibited atrophy mainly in the medial temporal and medial parietal areas^{3,5}.”

“The spreading pattern of cortical atrophy in ADCI typically first involves the medial temporal area, followed by the medial parietal, lateral temporal and lateral parietal, high-order sensory association, and prefrontal areas, and finally the primary sensorimotor areas.”

In the methods section,

“For the SuStaIn analysis, we considered five brain areas, including the medial temporal, inferior temporal, posterior medial parietal, lateral parietal, and frontal areas, based on prior studies and established clinical knowledge, prioritizing brain areas of notable clinical relevance and importance^{3,5}.”

In the discussion section,

“We selected the five sub-parcellations based on prior studies and established clinical knowledge, prioritizing brain areas of notable clinical relevance and importance. However, given these sub-parcellations are so large, there is large physiological variance even within each sub-parcellation.”

4. Moreover, the choice of the specific patient subgroups (ADCI vs. SVCI) is not fully clear. Please elaborate on the benefit of investigating them next to each other in one study.

Response: Both ADCI and SVCI are two major contributors to dementia with insidious onset and gradual progression. They also share overlapping pathology in some cases: concurrent CSVD burden is more frequently observed in ADCI compared to other neurodegenerative diseases. Likewise, amyloid markers are reported to be positive in 30-53% of SVCI patients. Thus, considering that ADCI and SVCI showed different distribution patterns of brain atrophy through different mechanisms, understanding the atrophic subtypes and stages for these two diseases could provide insights into their respective pathomechanisms and aid in their differential diagnosis. For example, the pattern of cortical atrophy in SVCI patients with mixed AD pathology may reflect that of AD, potentially exhibiting medial temporal and parietal thinning.

In response to the reviewer’s comments, we addressed this issue in our revised manuscript as

follows.

In the introduction section,

“ADCI and SVCI also share overlapping pathology in some cases: concurrent CSVD burden is more frequently observed in ADCI compared to other neurodegenerative diseases. Likewise, amyloid markers are reported to be positive in 30-53% of SVCI patients. Thus, considering that ADCI and SVCI showed different distribution patterns of brain atrophy through different mechanisms, understanding the atrophic subtypes and stages for these two diseases could provide insights into their respective pathomechanisms and aid in their differential diagnosis. For example, the pattern of cortical atrophy in SVCI patients with mixed AD pathology may reflect that of AD, potentially exhibiting medial temporal and parietal thinning.”

5. Table 1 does not list demographic details of the CU population. Please provide those since that information is essential to understand whether the control population is an adequate reference for the patient populations.

Response: In response to the reviewer’s comments, we added the information of the CU population in Table 1. There were differences in age, sex, and education level between both groups (CU vs. ADCI and CU vs. SVCI). Therefore, we considered w-scores (i.e., z-scores adjusted for co-variates) of cortical thickness controlling for age, sex, and education level. In response to the reviewer’s comments, we addressed this issue in our revised manuscript as follows.

	Total (n = 1,340)	CU (n=419)	ADCI (n = 713)	SVCI (n = 208)	p ADCI vs. SVCI
Demographics					
Age		69.4 ± 8.1	70.8 ± 9.3*	77.0 ± 7.8*	< 0.001
Sex, female (n (%))		252 (60.1%)	404 (56.7%)*	148 (71.2%)*	< 0.001
Education (years)		12.1 ± 4.7	11.7 ± 4.7*	8.7 ± 5.4*	< 0.001
APOE genotype					< 0.001
E3 homozygotes		297 (70.9%)	281 (39.4%)	137 (65.9%)	
E2 carrier		41 (9.8%)	21 (3.0%)	16 (7.7%)	
E4 carrier		81 (19.3%)	411 (57.6%)	55 (26.4%)	
Amyloid PET positivity (n (%))		0	713 (100.0%)	71 (34.1%)	< 0.001
Cognitive profiles					
K-MMSE		28.2 ± 1.9	21.8 ± 6.2	21.8 ± 5.3	1.000 [†]
SNSB-II		(n=369)	(n = 554)	(n = 162)	
Digit span backward		4.2 ± 1.4	3.4 ± 1.3	3.0 ± 1.0	0.304 [†]
K-BNT		48.9 ± 6.6	38.7 ± 12.9	35.9 ± 11.1	1.000 [†]
RCFT-copy		32.8 ± 3.5	25.7 ± 10.5	24.3 ± 9.0	0.888 [†]
SVLT-delayed recall		7.0 ± 2.2	1.6 ± 2.4	2.4 ± 2.6	< 0.001 [†]
RCFT-delayed recall		15.2 ± 6.4	4.7 ± 5.7	6.3 ± 6.4	< 0.001 [†]

COWAT-Semantic	33.5 ± 9.4	22.3 ± 10.0	19.1 ± 9.7	0.568 [†]
Stroop color reading	88.8 ± 21.1	51.9 ± 32.7	42.4 ± 26.5	0.312 [†]

*Indicates a significant statistical difference at $p < 0.05$ between CU and ADCI and CU and SVCI for age, sex, and education level.

[†]The p values were obtained by analysis of covariance after controlling for age, sex, and education level, and adjusted using Bonferroni's method because of inflated type I error.

In the resesult section,

“Table 1 shows the demographics of the patients with CU, ADCI, and SVCI. There were differences in age, sex, and education level between group comparisons (CU vs. ADCI and CU vs. SVCI). Therefore, we considered w-scores after controlling for age, sex, and education level.”

6. Acquisition of MRI data (ll 114): The authors state to have acquired a “standardized T2-weighted, three-dimensional (3D) T1-weighted turbo field echo, three-dimensional fluid-attenuated inversion recovery” > here, there might be typo, (“T2-weighted [...] T1-w image”?)

Response: To avoid confusion, we deleted “T2-weighted” in our revised manuscript.

7. Ll 174: The authors state to have corrected CT for “age, gender and education”. It is not specified how the correction procedure was performed. However, this information is crucial to track the adequacy of the analysis for the study question (see comments above)

Response: Please see the previous response. We updated the manuscript as follows.

In the method section,

“For the SuStaIn analysis, we considered five brain areas, including the medial temporal, inferior temporal, posterior medial parietal, lateral parietal, and frontal areas, which are known to be associated with ADCI and SVCI. The mean thickness was computed for each region and adjusted for co-variates including age, sex, and education level. Initially, we assessed differences in age, sex, and education level between both groups (CU vs. ADCI and CU vs. SVCI). The corrected value was considered as the residuals derived from multivariate regression analysis between cortical thickness and age, sex, and education level. Finally, we transformed adjusted cortical thickness measurements into z-scores relative to the entire normal population (noted as w-scores)⁴¹.”

8. L 212: a “two-tailed p-value of 0.05” was considered statistically significant > this is a little confusing since usually, a two-tailed p-value is significant below $0.05/2 = 0.025$. Please comment/specify.

Response: We have corrected this error in the revised manuscript by stating, “p-value < 0.05 was considered statistically significant for all analyses”.

9. L 255: “Interactive effect” should read “interaction effect”

Response: We have corrected the typo to "interaction effect" in the revised manuscript.

10. L1 306: *Here, in the middle of the discussion, a new hypothesis appears (“we hypothesized that the parietal subtype might be related to SVCI in participants with mixed AD pathologies, because the parietal area is known to be more vulnerable to AD”) – which had never been articulated before in the manuscript. Please avoid confronting the reader with new hypothesis in the discussion. Either delete this discussion section, or preferable include this as a hypothesis into your introduction and specify methods / results at the adequate sections in the manuscript.*

Response: Thank you for your keen observation. We have taken your advice and added the hypothesis to the introduction section to provide a more coherent narrative throughout the manuscript as follows:

In the introduction section,

“ADCI and SVCI also share overlapping pathology in some cases: concurrent CSVD burden is more frequently observed in ADCI compared to other neurodegenerative diseases. Likewise, amyloid markers are reported to be positive in 30-53% of SVCI patients. Thus, considering that ADCI and SVCI showed different distribution patterns of brain atrophy through different mechanisms, understanding the atrophic subtypes and stages for these two diseases could provide insights into their respective pathomechanisms and aid in their differential diagnosis. For example, the pattern of cortical atrophy in SVCI patients with mixed AD pathology may reflect that of AD, potentially exhibiting medial temporal and parietal thinning.”

11. Tables, minor suggestion: *the asterisk symbol (“*”) is usually associated with emphasizing significant p-values. Here, you use it to specify how p-values were obtained. It could be helpful to some reader to use a different symbol or footnote and use the asterisk to emphasize p-value-significance instead.*

Response: To avoid confusion, we have revised the tables and adopted a different symbol (†) to specify how p-values were obtained.

References

- 1 Galasko, D., Abramson, I., Corey-Bloom, J. & Thal, L. J. Repeated exposure to the Mini-Mental State Examination and the Information-Memory-Concentration Test results in a practice effect in Alzheimer's disease. *Neurology* **43**, 1559-1563, doi:10.1212/wnl.43.8.1559 (1993).
- 2 La Joie, R. *et al.* Region-specific hierarchy between atrophy, hypometabolism, and β -amyloid (A β) load in Alzheimer's disease dementia. *J Neurosci* **32**, 16265-16273, doi:10.1523/jneurosci.2170-12.2012 (2012).
- 3 Kim, C. H. *et al.* Cortical thinning in subcortical vascular dementia with negative 11C-PiB PET. *J Alzheimers Dis* **31**, 315-323, doi:10.3233/JAD-2012-111832 (2012).
- 4 Noh, Y. *et al.* Anatomical heterogeneity of Alzheimer disease: based on cortical thickness on MRIs. *Neurology* **83**, 1936-1944, doi:10.1212/wnl.0000000000001003 (2014).
- 5 Seo, S. W. *et al.* Cortical thinning in vascular mild cognitive impairment and vascular dementia of subcortical type. *J Neuroimaging* **20**, 37-45, doi:10.1111/j.1552-6569.2008.00293.x (2010).

Reviewers' comments:

Reviewer #1 (Remarks to the Author):

Thank you for revising the manuscript according to my comments, and the revised manuscript seems delivering more clear data.

With rebuttal comments raised by the authors, I still got two major concerns: clustering analysis using the sustain methodology and cognitive decline per subtype. First, according to selection of the optimal number of subtypes, I cannot be convinced why using two subtypes should be optimal, especially for ADCI patients. To me, the log-likelihood analysis and CVIC analysis could not make significant difference across different numbers of subtypes. This might imply that the sustain method may not be an appropriate model for clustering the authors' data. By looking at the supplementary Fig 1, I'm quite doubtful why the authors employ the sustain model for clustering.

Second, Figure 3 shows longitudinal trajectory of K-MMSE scores in patients with ADCI and SVCI groups, while the trend of increasing K-MMSE scores over time in the diffuse subtype of ADCI is still confusing. The authors claimed that this confusing trend could be due to the fact that the K-MMSE being repeated at relatively short intervals, it seems there might have been a learning effect in these patients. If this somehow implies the measure of K-MMSE would not be stable, then this indeed raises bigger concerns since all the cognitive function of the patients in the current study was evaluated using the K-MMSE scores. The authors might be able to explain the reason with better logic, and I would like to suggest to break down the diffuse subtype more and look at smaller subgroup.

Reviewer #2 (Remarks to the Author):

Dear Editor,

thank you for the opportunity to review the revised version of the manuscript "Distinct spatiotemporal patterns of cortical thinning in Alzheimer's disease-type cognitive impairment and subcortical vascular cognitive impairment". The authors are now providing a thoroughly revised version which addresses all major concerns raised in my initial review. Specifically, the following issues were clarified / revised:

- Clarification of the suitability of the SUSTAIN model for spatiotemporal analyses (additional references and elaboration in the main text)
- Clarification of statistical methods and correction procedures; additional demographic data was provided (which was requested for the CU group)
- Restructuring of the hypotheses (previously, a new hypothesis had been raised in the discussion section – this was now corrected for)
- Clarification of the choice of ROIs (in text and with additional clinical references), as well as acknowledging limitations of those (rather big) ROIs in the discussion section
- Clarification of smaller inaccuracies in text / tables

In light of these revisions, I'm happy to provide my recommendation to accept this article for publication.

Thank you again for considering me as a reviewer for this interesting manuscript.

Sincerely,
Marlene Tahedl.

Response to the Reviewers
REVIEWER 1

First, according to selection of the optimal number of subtypes, I cannot be convinced why using two subtypes should be optimal, especially for ADCI patients. To me, the log-likelihood analysis and CVIC analysis could not make significant difference across different numbers of subtypes. This might imply that the sustain method may not be an appropriate model for clustering the authors' data. By looking at the supplementary Fig 1, I'm quite doubtful why the authors employ the sustain model for clustering.

Response: We agree that there is no statistically significant difference in log-likelihood values between using two and three subtypes for ADCI patients. Thus, we report clustering ADCI patients into three subgroups. Comparing the results of using two and three subtypes (see Supplementary Fig 2) for the ADCI patients, we noted similarities in visual profiles. That is, the medial temporal subtype of using two clusters corresponded to subtypes 2 and 3 of using three clusters, while the diffusion subtype of using two clusters matched with subtype 1 of using three clusters. Hence, we compared the clinical scores of the medial temporal subtype (2 clusters) with those of subtypes 2 and 3 (3 clusters) and found no significant differences (Supplementary Table 1). We also compared the clinical scores of the diffusion subtype (2 clusters) with those of subtype 1 (3 clusters) and found minor differences (i.e., 2 scores different out of 12 scores). In sum, the subtypes of using two clusters and those of using three clusters had similarities in clinical scores. Since the results of using two and three clusters had similarities, we report the results of two clusters in the main manuscript due to their succinctness and the results of three clusters in the supplement. The SustaIn results of using three subtypes are added in the supplement (Supplementary Table 2 and 3).

We further revised the manuscript as follows:

In the result section,

“The optimal number of subtypes was determined by using CVIC and log-likelihood with respect to varying number of subtypes (Supplementary Fig.1). *Using two subtypes had the lowest CVIC and highest log-likelihood in the SVCI patients. Using two subtypes had the lowest CVIC, but there was no statistically significant difference in log-likelihood values between using two and three subtypes for ADCI patients. Additional results of using three subtypes and the comparison with using two subtypes for the ADCI patients are reported in the supplement (Supplementary Fig. 2 and Table. 1-3).*”

Sustain result for 2 class subtypes

Sustain result for 3 class subtypes

Supplementary Fig 2. Comparison biomarkers for using two (top row) and three (bottom) subtypes for the ADCI patients. The medial temporal subtype of the top row corresponded to subtypes 2 and 3 in the bottom row, while the diffusion subtype of the top row matched with subtype 1 in the bottom row.

Supplementary Table 1. Comparison of subtypes between using two and three clusters for the ADCI patients.

Group comparison for SustaIn subtypes	Medial temporal subtypes			p	Diffuse subtypes		p
	2 class subtype1 n = 469	3 class subtype2 n = 234	3 class subtype3 n = 187		2 class subtype2 n = 244	3 class subtype1 n = 292	
Demographics							
Age	70.3 ± 9.4	71.0 ± 9.4	68.9 ± 8.9	0.090 [†]	71.7 ± 9.2	70.9 ± 8.8	0.738 [†]
Female sex (n (%))	227 (59.1%)	129 (55.1%)	99 (52.9%)	0.305 [†]	127 (52.1%)	176 (60.3%)	0.056 [†]
Education (years)	11.6 ± 4.6	11.6 ± 4.9	11.7 ± 4.6	0.983 [†]	11.8 ± 4.8	11.7 ± 4.5	0.906 [†]
APOE genotypes				0.832 [†]			0.674 [†]
E3 homozygotes	186 (39.7%)	91 (38.9%)	71 (38.0%)		95 (38.9%)	91 (40.8%)	
E2 carrier	13 (2.8%)	6 (2.6%)	2 (1.1%)		8 (3.3%)	6 (4.4%)	
E3 carrier	270 (57.6%)	137 (58.5%)	114 (60.9%)		141 (57.8%)	137 (54.8%)	
Cognitive profiles							
K-MMSE	21.4 ± 6.2	22.7 ± 5.6	22.5 ± 6.1	1.000 [‡]	22.5 ± 6.1	22.0 ± 5.6	0.568 [‡]
SNSB-II	(n=366)	(n=183)	(n=138)		(n=188)	(n=223)	
Digit span backward	3.5 ± 1.4	3.6 ± 1.4	3.4 ± 1.5	1.000 [‡]	3.4 ± 1.2	3.1 ± 1.2	1.000 [‡]
K-BNT	37.9 ± 13.3	38.6 ± 12.5	37.1 ± 15.0	1.000 [‡]	38.9 ± 13.0	37.7 ± 13.1	1.000 [‡]
RCFT-copy	25.7 ± 10.5	26.5 ± 10.2	24.2 ± 11.6	1.000 [‡]	24.7 ± 11.3	24.4 ± 10.9	1.000 [‡]
SVLT-delayed recall	1.3 ± 2.1	1.4 ± 2.3	1.9 ± 2.6	0.206 [‡]	2.0 ± 2.7	1.4 ± 2.2	0.016 [‡]
RCFT-delayed recall	4.1 ± 5.3	4.2 ± 5.4	5.8 ± 6.5	0.094 [‡]	5.7 ± 6.4	4.0 ± 5.2	0.040 [‡]
COWAT-Semantic	21.8 ± 10.1	21.6 ± 10.2	23.0 ± 11.5	1.000 [‡]	23.2 ± 9.9	20.9 ± 9.3	0.544 [‡]
Stroop color reading	52.0 ± 32.5	56.7 ± 32.5	54.0 ± 32.5	1.000 [‡]	51.7 ± 33.2	47.3 ± 31.5	0.360 [‡]

[†]The p values were adjusted using Bonferroni's method because of inflated type I error.

[‡]The p values were obtained by analysis of covariance after controlling for age, sex, and education level, and adjusted using Bonferroni's method because of inflated type I error.

ADCI = Alzheimer's disease-type cognitive impairment; APOE = Apolipoprotein; K-MMSE = Korean version of the Mini-Mental State Examination; SNSB-II = Seoul Neuropsychological Screening Battery 2nd edition; K-BNT = Korean version of the Boston Naming Test; RCFT = Rey–Osterrieth Complex Figure Test; SVLT = Seoul Verbal Learning Test; COWAT = Controlled Oral Word Association Test

Supplementary Table 2. Baseline demographic variables and cognitive profiles of each SuStAI subtypes using three clusters in ADCI patients.

ADCI (n = 713)	Diffuse-like			Medial temporal-like	
	Subtype 1 (n=292)	Subtype 2 (n = 234)	Subtype 3 (n = 187)	p	post hoc analysis
Demographics					
Age	71.2±9.2	71.4±9.6	69.2±9.0	0.108 [†]	
Sex, female (n (%))	176 (60.3)	129 (55.1)	99 (52.9)	0.729 [†]	
Education (years)	71.2±9.2	71.4±9.6	69.2±9.0	1.000 [†]	
APOE genotype				0.492 [†]	
E3 homozygotes	121 (41.4)	91 (38.9)	69 (36.9)		
E2 carrier	13 (4.5)	6 (2.6)	2 (1.1)		
E4 carrier	158 (54.1)	137 (58.6)	116 (62.0)		
Cognitive profiles					
K-MMSE	21.7±5.9	22.0±6.1	21.7±6.9	0.800 [‡]	
SNSB-II	(n=227)	(n = 220)	(n = 177)		
Digit span backward	3.1±1.2	3.5±1.5	3.3±1.5	0.011 [‡]	2>1
K-BNT	37.2±13.2	37.5±13.6	35.3±16.3	0.074 [‡]	
RCFT-copy	24.0±11.1	26.0±10.5	23.4±12.1	0.056 [‡]	
SVLT-delayed recall	1.3±2.2	1.4±2.3	1.8±2.5	0.054 [‡]	
RCFT-delayed recall	3.9±5.1	4.0±5.3	5.3±6.4	0.011 [‡]	
COWAT-Semantic	20.8±9.5	21.4±10.3	22.8±11.5	0.169 [‡]	
Stroop color reading	44.9±31.7	53.3±34.0	48.9±34.6	0.008 [‡]	2>1

[†]The *p* values were adjusted using Bonferroni's method because of inflated type I error.

[‡]The *p* values were obtained by analysis of covariance after controlling for age, sex, and education level, and adjusted using Bonferroni's method because of inflated type I error.

ADCI = Alzheimer's disease-type cognitive impairment; APOE = Apolipoprotein; K-MMSE = Korean version of the Mini-Mental State Examination; SNSB-II = Seoul Neuropsychological Screening Battery 2nd edition; K-BNT = Korean version of the Boston Naming Test; RCFT = Rey–Osterrieth Complex Figure Test; SVLT = Seoul Verbal Learning Test; COWAT = Controlled Oral Word Association Test

Supplementary Table 3. Mixed effects models analyzing the relationship between SuStaIn subtypes using three clusters and rate of cognitive decline in ADCI patients

Cognition parameter	ADCI subtype × time		post hoc analysis
	β (S.E)	p for interaction	
K-MMSE			
2 vs 1	-0.349 (0.07)	< 0.001 [†]	
3 vs 1	-0.536 (0.09)	< 0.001 [†]	2=3>1
3 vs 2	-0.185 (0.09)	0.984 [†]	
SNSB-II			
Digit span backward			
2 vs 1	-0.077 (0.02)	< 0.001 [†]	
3 vs 1	-0.065 (0.03)	0.408 [†]	2>1
3 vs 2	0.028 (0.03)	1.000 [†]	
K-BNT			
2 vs 1	-0.593 (0.14)	< 0.001 [†]	
3 vs 1	-0.743 (0.18)	< 0.001 [†]	2=3>1
3 vs 2	0.023 (0.21)	1.000 [†]	
RCFT-copy			
2 vs 1	-0.179 (0.13)	1.000 [†]	
3 vs 1	-0.374 (0.17)	0.648 [†]	
3 vs 2	-0.136 (0.18)	1.000 [†]	
SVLT-delayed recall			
2 vs 1	-0.103 (0.04)	0.264 [†]	
3 vs 1	-0.030 (0.05)	1.000 [†]	
3 vs 2	0.068 (0.06)	1.000 [†]	
RCFT-delayed recall			
2 vs 1	-0.250 (0.09)	1.000 [†]	
3 vs 1	-0.166 (0.12)	1.000 [†]	
3 vs 2	0.127 (0.13)	1.000 [†]	
COWAT-Semantic			
2 vs 1	-0.486 (0.16)	0.048 [†]	
3 vs 1	-0.904 (0.20)	< 0.001 [†]	2=3>1
3 vs 2	-0.333 (0.22)	1.000 [†]	
Stroop color reading			
2 vs 1	-0.702 (0.41)	1.000 [†]	
3 vs 1	-1.275 (0.51)	0.312 [†]	
3 vs 2	-0.386 (0.57)	1.000 [†]	

Model: fixed effect: age, sex, education level, SuStaIn subtype, baseline cognition, time interval from initial evaluation (years), sustain subtype × time interval; random effect: subject.

[†]*p* values were adjusted using Bonferroni's method because of inflated type I error.

Three subtypes were indicated as follows: 1, diffuse-like subtype; 2 and 3, medial temporal-like subtypes.

ADCI = Alzheimer's disease-type cognitive impairment; APOE = Apolipoprotein; K-MMSE = Korean version of the Mini-Mental State Examination; SNSB-II = Seoul Neuropsychological Screening Battery 2nd edition; K-BNT = Korean version of the Boston Naming Test; RCFT = Rey–Osterrieth Complex Figure Test; SVLT = Seoul Verbal Learning Test; COWAT = Controlled Oral Word Association Test;

Second, Figure 3 shows longitudinal trajectory of K-MMSE scores in patients with ADCI and SVCI groups, while the trend of increasing K-MMSE scores over time in the diffuse subtype of ADCI is still confusing. The authors claimed that this confusing trend could be due to the fact that the K-MMSE being repeated at relatively short intervals, it seems there might have been a learning effect in these patients. If this somehow implies the measure of K-MMSE would not be stable, then this indeed raises bigger concerns since all the cognitive function of the patients in the current study was evaluated using the K-MMSE scores. The authors might be able to explain the reason with better logic, and I would like to suggest to break down the diffuse subtype more and look at smaller subgroup.

Response: We appreciate the opportunity to clarify the longitudinal trajectory of K-MMSE scores in patients with ADCI, particularly focusing on the diffuse subtype. In response to the reviewer's comments, we investigated the distribution of cognitive stages in our participants. Our detailed analysis revealed that the frequency of preclinical AD ($A\beta(+)$ cognitively unimpaired (CU) individuals) is significantly higher in the diffuse subtype of ADCI compared to the medial temporal subtype (18.6% vs. 7.6%, $p=0.001$). This would be an important consideration, as previous studies investigating the trajectories of preclinical AD have shown that only 13.5%–22.9% of these patients progress to cognitive decline¹⁻³. One study reported that 65.5% of preclinical AD patients either maintain stable cognitive functions or even experience a slight improvement.⁴

Thus, we addressed this issue in the revised manuscript as follows:

In the discussion section,

“... We selected the five sub-parcellations based on prior studies and established clinical knowledge, prioritizing brain areas of notable clinical relevance and importance. However, given these sub-parcellations are so large, there is a large physiological variance even within each sub-parcellation. Additionally, we observed an increasing trend of K-MMSE scores in the diffuse subtype of ADCI. This trend may reflect differences in the distribution of cognitive stages in each subtype. Notably, the diffuse subtype exhibits a significantly higher frequency of preclinical AD ($A\beta$ -positive CU individuals) compared to the medial temporal subtype (18.6% vs. 7.6%, $p=0.001$). This would be an important consideration, as previous studies investigating the trajectories of preclinical AD have shown that only 13.5%–22.9% of these patients progress to cognitive decline⁵⁰⁻⁵². One study reported that 65.5% of preclinical AD patients either maintain stable cognitive functions or even experience a slight improvement⁵³.”

- 1 Knopman, D. S. *et al.* Short-term clinical outcomes for stages of NIA-AA preclinical Alzheimer disease. *Neurology* **78**, 1576-1582, doi:10.1212/WNL.0b013e3182563bbe (2012).
- 2 Lim, Y. Y. *et al.* APOE ϵ 4 moderates amyloid-related memory decline in preclinical Alzheimer's disease. *Neurobiol Aging* **36**, 1239-1244, doi:10.1016/j.neurobiolaging.2014.12.008 (2015).
- 3 Toledo, J. B. *et al.* Neuronal injury biomarkers and prognosis in ADNI subjects with normal cognition. *Acta Neuropathol Commun* **2**, 26, doi:10.1186/2051-5960-2-26 (2014).
- 4 Pietrzak, R. H. *et al.* Trajectories of memory decline in preclinical Alzheimer's disease: results from the Australian Imaging, Biomarkers and Lifestyle Flagship Study of ageing. *Neurobiol Aging* **36**, 1231-1238, doi:10.1016/j.neurobiolaging.2014.12.015 (2015).

REVIEWERS' COMMENTS:

Reviewer #1 (Remarks to the Author):

Thanks for the revision according to the review comments. I can now confirm that the proposed manuscript is appropriate for the publication from communications in biology.